# Adaptively Secure Efficient (H)IBE over Ideal Lattice with Short Parameters

**DOI:** 10.3390/e22111247

**Published:** 2020-11-02

**Authors:** Yuan Zhang, Yuan Liu, Yurong Guo, Shihui Zheng, Licheng Wang

**Affiliations:** State Key Laboratory of Networking and Switching Technology, Beijing University of Posts and Telecommunications, Beijing 100876, China; zy_bupt@bupt.edu.cn (Y.Z.); yuanl_1011@163.com (Y.L.); gyr_bupt@163.com (Y.G.); shihuizh@bupt.edu.cn (S.Z.)

**Keywords:** lattice, IBE, adaptive security, short parameter, standard model, RLWE

## Abstract

Identity-based encryption (IBE), and its hierarchical extension (HIBE), are interesting cryptographic primitives that aim at the implicit authentication on the users’ public keys by using users’ identities directly. During the past several decades, numerous elegant pairing-based (H)IBE schemes were proposed. However, most pairing-related security assumptions suffer from known quantum algorithmic attacks. Therefore, the construction of lattice-based (H)IBE became one of the hot directions in recent years. In the setting of most existing lattice-based (H)IBE schemes, each bit of a user’s identity is always associated with a parameter matrix. This always leads to drastic but unfavorable increases in the sizes of the system public parameters. To overcome this issue, we propose a flexible trade-off mechanism between the size of the public parameters and the involved computational cost using the blocking technique. More specifically, we divide an identity into $l^{\prime }$ segments and associate each segment with a matrix, while increasing the lattice modulo slightly for maintaining the same security level. As a result, for the setting of 160-bit identities, we show that the size of the public parameters can be reduced by almost 89.7% (resp. 93.8%) while increasing the computational cost by merely 5.2% (resp. 12.25%) when $l^{\prime }$ is a set of 16 (resp. 8). Finally, our IBE scheme is extended to an HIBE scheme, and both of them are proved to achieve the indistinguishability of ciphertexts against adaptively chosen identity and chosen plaintext attack (IND-ID-CPA) in the standard model, assuming that the well-known ring learning with error (RLWE) problem over the involved ideal lattices is intractable, even in the post-quantum era.

## 1. Introduction

Identity-based encryption (IBE), first introduced by Shamir [1], is an interesting public-key encryption mechanism. It reduces the complexity of system and the cost of establishing public-key infrastructure. The public keys are users’ identities directly, and the corresponding private keys can only be generated by the private-key generator (PKG). Moreover, IBEs can be used for confidential communication, network protocols, digital signatures, etc. In 2001, Boneh and Franklin [2] constructed the first practical IBE scheme under the bilinear Diffe–Hellman (BDH) assumption. Then, Canetti et al. [3] constructed an IBE scheme in the standard model, and they gave the security proof in the selective-ID model. In this model, the adversary must announce the target identity at the beginning. Boneh and Boyen [4] proposed a fully (adaptively) secure IBE scheme. Their scheme is too inefficient to be practical since it requires numerous exponentiation operations and group operations. In the adaptive-ID model, the adversary can announce the target identity after private key queries. In 2005, Waters [5] constructed the first efficient fully secure IBE scheme and showed that a selectively secure scheme can be improved to adaptive security. Furthermore, there are many IBE constructions [6,7,8,9,10,11,12,13] based on pairing or quadratic residues which cannot resist quantum computing.

Lattice-based cryptography has become the focus of research in recent years because it is flexible in construction and resistant to quantum computing. Regev [14] defined the learning with error (LWE) problem and gave a reduction from the worst-case lattice problems. Stehlé [15] and Lyubashevsky [16] defined the ring learning with error (RLWE) problem, which led to new cryptographic applications.

In 2008, Gentry et al. [17] proposed the first LWE-based IBE scheme in the random oracle model. Their scheme relied on the Dual-Regev encryption scheme and became an example of an LWE-based IBE scheme. Agrawal et al. [18] then construct an efficient selectively secure IBE scheme based on LWE problem in the standard model. They also give an adaptively secure IBE scheme, but each bit of a user’s identity is associated with a parameter matrix. This always leads to drastic but unfavorable increases in the sizes of the system public parameters. To solve this drawback, Singh et al. [19] constructed efficient adaptively secure (hierarchical) IBE schemes with short parameters using the blocking technique [20,21]. In 2016, Yamada [22] constructed an adaptively secure IBE scheme with short parameters using injective map and homomorphic computation. Zhang et al. [23] proposed an adaptively secure IBE scheme which achieved shorter public parameters, but their scheme only achieved Q-bounded security. In 2017, Yamada [24] constructed new adaptively secure IBE schemes via new partitioning functions, but the public parameters in their scheme are larger than [23]. Moreover, there are many other IBE constructions [25,26,27,28,29,30,31,32] based on the LWE problem.

Compared with the LWE problem, the RLWE problem is more practical in construction because of smaller storage and faster calculation. In particular, we can use fast Fourier transform (FFT) or number theoretic transform (NTT) to accelerate polynomial multiplications. In 2013, Yang et al. [33] construct a selectively secure IBE scheme over ideal lattice in the standard model. Their construction is a ring variant of Agrawal’s selective-ID scheme [18]. In 2014, Ducas et al. [34] propose an efficient IBE scheme over Number Theory Research Unit (NTRU) lattice. (NTRU is a ring-based public key cryptosystem, which was proposed by Hoffstein [35] in 1998. The lattice specified in their scheme is often called the NTRU lattice.) Their construction is a NTRU variant of the scheme by [17]. In order to achieve shorter public parameters, Katsumata [36] constructs an adaptively secure IBE scheme over ideal lattice using Yamada’s method [22]. In 2018, Bert et al. [37] construct an efficient IBE scheme and give an efficient implementation. Their construction uses the ring-version trapdoor of Micciancio [38] which is efficient and easy to implement. However, their scheme only achieves selective security. Therefore, it is meaningful to construct adaptively secure efficient (H)IBE schemes over ideal lattice with shorter parameters.

Our contribution. In this paper, we first construct an adaptively secure IBE scheme over ideal lattice with short parameters. In the setting of the most existing lattice-based (H)IBE schemes, the public parameters are generally composed of l+2 matrices, where *l* is the bit length of user’s identity. Using the blocking technique, we can reduce the number of elements in public parameters from l+2 to l/β+2 where β is a flexible constant. However, this leads to a reduction in the security. We need to increase the lattice modulo *q* to achieve the same security level as [18], but it causes an increase in computational cost. Therefore, we make a trade-off between storage space and computational cost. For l=160, the size of public parameters can be reduced by almost 89.7% while increasing the computational cost by only 5.2%. When β is set of 20 (resp. 10), the public parameters only contain 10 (resp. 18) vectors. According to our performance analysis, our scheme can achieve shorter public parameters and better computational efficiency. In addition, we use the gadget-based trapdoor as [37,38] which is simple, efficient and smaller in storage than a basis. Finally, we extend our IBE scheme to a hierarchical IBE scheme, and both of them are proved achieving the indistinguishability of ciphertexts against adaptively chosen identity and chosen plaintext attack (IND-ID-CPA) in the standard model.

The rest of this paper is organized as follows. Section 2 is preliminaries. Section 3 and Section 4 describe our adaptively secure IBE and HIBE schemes. In Section 5, we analyse the trade-off and compare with other constructions. In Section 6, we summarize this paper.

## 2. Preliminaries

Notation. In this paper, we use uppercase letters to represent matrix (i.e., *A*), and lowercase letters to represent constant or polynomial (i.e., *l* or *u*). We use uppercase bold letters to represent polynomial matrices (i.e., ***R***), and lowercase bold letters to represent polynomial vectors (i.e., ***a***). We use negligible function to represent the function ϵ(n) which is less than all polynomial fractions for sufficiently large *n*. We use overwhelming probability to indicate that the event happens with probability 1−ϵ(n).

### 2.1. IBE and Hierarchical IBE

HIBE system contains four algorithms [7,8]. For identity $id=(id_{1},\cdots ,id_{l})$, we describe the HIBE system as follows.

**Setup**(d,λ): On input a security parameter λ and a maximum depth *d*, the algorithm outputs the public parameters PP and master key MK.

**Derive**$\left(PP,id|id_{l},SK_{id|id_{l-1}},MK\right)$: On input public parameters PP, master key MK, identity $id|id_{l}$ at depth *l*, and private key $SK_{id|id_{l-1}}$ at depth l−1, it outputs the private key $SK_{id|id_{l}}$ at depth *l*.

**Encrypt**$\left(PP,\mu ,id|id_{l}\right)$: On input public parameters PP, an identity $id|id_{l}$ at depth *l* and a message μ, the algorithm outputs a ciphertext CT.

**Decrypt**$\left(PP,CT,SK_{id|id_{l}}\right)$: On input public parameters PP, a ciphertext CT and a private key $SK_{id|id_{l}}$, the algorithm outputs the message μ.

IBE system is the same as above HIBE system when d=1. Compared with HIBE, there is an algorithm Extract instead of algorithm Derive. The algorithm Extract inputs public parameters PP, identity id, master key MK, and it outputs the corresponding private key $SK_{id}$.

**Security Game.** We use an indistinguishable from random game to define the adaptive security of (H)IBE, which means that adversary can not distinguish between challenge ciphertext and random ciphertext. Let $M_{\lambda }$ and $C_{\lambda }$ be the message space and ciphertext space where λ is a security parameter. For a maximum depth *d*, the following defines the game.

**Setup**: The challenger runs algorithm Setup(d,λ) and sends the public parameters PP to the adversary.

**Phase 1**: The adversary performs private key queries $q_{1},\cdots ,q_{m},$ and the event $q_{i}$ corresponds to the identity $id_{i}$. The challenger runs algorithm Extract to generate the private key $sk_{i}$ corresponding to $id_{i}$ and sends it to the adversary.

**Challenge**: The adversary submits a plaintext $M\in M_{\lambda }$ and a target identity $id^{*}$ which can not appear in Phase 1. Then the challenger chooses a random bit r∈{0,1} and a random ciphertext $C\in C_{\lambda }$. If r=0, the challenger sets the challenge ciphertext $C^{*}:=$ Encrypt $\left(PP,M,id^{*}\right)$. Otherwise, it sets the challenge ciphertext $C^{*}=C$. The challenger sends $C^{*}$ to the adversary.

**Phase 2**: The adversary performs adaptive queries $q_{m+1},\cdots ,q_{n}$. The event $q_{i}$ corresponds to the identity $id_{i}$ which can not be $id^{*}$. The challenger responds as in Phase 1.

**Guess**: The adversary outputs a guess $r^{\prime }\in \left\{0,1\right\},$ and wins if $r^{\prime }=r$.

The adversary A described above is a IND-ID-CPA attacker. We define the advantage of A as
$$Adv_{d,\varepsilon ,A}\left(\lambda \right)=\left|Pr\left[r^{\prime }=r\right]-\frac{1}{2}\right|$$


**Definition** **1.**
*If for all IND-ID-CPA attackers A, the advantage $Adv_{d,\varepsilon ,A}\left(\lambda \right)$ is a negligible function, then the HIBE scheme ε is IND-ID-CPA security. The security model of IBE is the same as above model with d=1.*


The following Definition 2 defines the abort-resistant hash functions [18,19], which is used in our security proof.

**Definition** **2**([18,19])**.**
*Let H:={H:X→Y} be a family of hash functions and Y contains element 0. For a set $\bar{x}=\left\{x_{0},x_{1},\cdots ,x_{Q}\right\}\in X^{Q+1}$ with $x_{0}\notin \left\{x_{1},\cdots ,x_{Q}\right\}$, we define the non-abort probability of $\bar{x}$*
$$\alpha \left(\bar{x}\right):=Pr\left[H\left(x_{0}\right)=0\wedge H\left(x_{1}\right)\neq 0\wedge \cdots \wedge H\left(x_{Q}\right)\neq 0\right]$$
*where the probability is over the random choice of H in H. For $\alpha \left(\bar{x}\right)\in \left[\alpha_{min},\alpha_{max}\right]$, the hash family H is $\left(Q,\alpha_{min},\alpha_{max}\right)$ abort-resistant.*


We use the abort-resistant hash family similar to [5,18]. Let *q* be a prime and ${\left(Z_{q}^{l^{\prime }}\right)}^{*}:=Z_{q}^{l^{\prime }}\backslash \left\{0^{l^{\prime }}\right\}$; we define the hash family $H_{\mathrm{Wat}}:\left\{H_{h}:{\left(Z_{q}^{l^{\prime }}\right)}^{*}\to Z_{q}\right\}$ as $H_{h}\left(id\right):=1+\sum_{i=1}^{l^{\prime }}h_{i}b_{i}\in Z_{q}$ where $id=\left(b_{1},\ldots ,b_{l^{\prime }}\right)\in {\left(Z_{q}^{l^{\prime }}\right)}^{*}$ and $h=\left(h_{1},\ldots ,h_{l^{\prime }}\right)\in Z_{q}^{l^{\prime }}$.

### 2.2. Integer Lattice and Ideal Lattice

**Definition** **3.**
*Let q be a prime, $A\in Z_{q}^{n\times m}$ and $u\in Z_{q}^{n}$; we define integer lattice as:*
$$\begin{array}{c} \Lambda_{q}\left(A\right):=\left\{e\in Z^{m}\;s.t.\;\exists s\in Z_{q}^{n}\;where\;A^{⊤}s=e\;\;\mathrm{mod}\;q\right\}\\ \Lambda_{q}^{⊥}\left(A\right):=\left\{e\in Z^{m}\;s.t.\;Ae=0\;\;\mathrm{mod}\;q\right\}\\ \Lambda_{q}^{u}\left(A\right):=\left\{e\in Z^{m}\;s.t.\;Ae=u\;\;\mathrm{mod}\;q\right\} \end{array}$$


**Ideal Lattice.** Let *n* be a power of 2; we define the modular polynomial $f\left(x\right)=x^{n}+1$. Then, we define the ring polynomial *R* as R=Z[x]/f(x). For a modulus *q*, we define the ring polynomial $R_{q}$ as $R_{q}=Z_{q}\left[x\right]/f\left(x\right)$. Therefore, elements in $R_{q}$ are polynomials with coefficients less than *q*. The following definition from [16,37] defines the Decision RLWE problem.

**Definition** **4**(Decision RLWE)**.**
*Given a vector of m uniformly random polynomials $a={\left(a_{1},\cdots ,a_{m}\right)}^{⊤}\in R_{q}^{m}$, and b=as+e where $s\in R_{q}$ and $e\in D_{R^{m},\sigma }$. Then, distinguish (a,b=as+e) from uniform (a,b).*

Similar to [18], we use $∥\tilde{S}∥$ to denote the Gram–Schmidt norm of *S* where $S=\{s_{1},\cdots ,s_{k}\}$ in $R^{m}$. We use $D_{L,\sigma ,c}$ to denote the discrete Gaussian distribution with center *c* and parameter σ over a set *L*. Moreover, the following theorem from [18,39] defines an algorithm ExtendBasis which is used in our HIBE construction.

**Theorem** **1**([18,39])**.**
*Let $A_{i}\in Z_{q}^{n\times m_{i}}$ where i=1,2,3, and $A:=(A_{1}|A_{2}|A_{3})$. We define the algorithm $ExtendBasis(A_{1},A_{2},A_{3},T_{2})$ which outputs a basis $T_{A}$ of $\Lambda_{q}^{⊥}\left(A\right)$ where $T_{2}$ is a basis of $\Lambda_{q}^{⊥}\left(A_{2}\right)$.*

### 2.3. Trapdoors on Lattice

Our constructions require the notion of trapdoor which is first introduced by Ajtai [40]. For a short basis $T_{A}$ of $\Lambda_{q}^{⊥}\left(A\right)$, we can get short vectors in $\Lambda_{q}^{⊥}\left(A\right)$ from a Gaussian distribution. We use the g-trapdoor introduced by Micciancio [38] and the following definition from [37] defines the ring variant of the g-trapdoor.

**Definition** **5**(g-trapdoor)**.**
*For $k=\lceil \log_{2}q\rceil$, m>k, let a be a vector in $R_{q}^{m}$ and g be a vector in $R_{q}^{k}$. The g-trapdoor for a is a polynomial matrix $T_{a}$ in $R^{(m-k)\times k}$ following a discrete Gaussian distribution of parameter σ, and satisfying $a^{⊤}\left(\begin{array}{c} \begin{array}{c} T_{a}\\ I_{k} \end{array} \end{array}\right)=hg^{⊤}$ for some invertible element $h\in R_{q}$. The polynomial h is the tag associated to trapdoor $T_{a}$.*

In our construction, we need a trapdoor generation algorithm (TrapGen) and preimage sampling algorithm (SamplePre) from [37], and both of them are described as follows.

Algorithm TrapGen inputs a modulus *q*, a Gaussian parameter σ, a polynomial vector $a^{\prime }\in R_{q}^{m-k}$ and a polynomial $h\in R_{q}$. It returns a polynomial vector $a\in R_{q}^{m}$, a trapdoor $T_{a}\in R^{(m-k)\times k}$ with tag *h*. We use vector $a^{\prime }$, gadget vector g and trapdoor $T_{a}$ to construct the target vector a. The trapdoor $T_{a}$ is choosing from a gaussian distribution with parameter σ. In our construction, the target vector a is part of public parameter and the trapdoor $T_{a}$ is the master key.

Algorithm SamplePre inputs a vector $a\in R_{q}^{m}$, a trapdoor $T_{a}\in R^{(m-k)\times k}$ with tag $h\in R_{q}$, a polynomial $u\in R_{q}$ and a Gaussian parameter σ. It returns a vector $x\in R_{q}^{m}$ following a discrete Gaussian distribution of parameter ξ, and satisfying $a^{⊤}x=u$. To find a vector x satisfing $a^{⊤}x=u$, we need to find a vector z that satisfies $g^{⊤}z=h^{-1}\cdot \left(u-a^{⊤}p\right)$ where p is a perturbation vector. Then, we get $x=p+(\begin{array}{c} \begin{array}{c} T_{a}\\ I_{k} \end{array} \end{array})z$ such that $a^{⊤}x=a^{⊤}p+a^{⊤}\left(\begin{array}{c} \begin{array}{c} T_{a}\\ I_{k} \end{array} \end{array}\right)z=a^{⊤}p+hg^{⊤}z=a^{⊤}p+h\cdot h^{-1}\left(u-a^{⊤}p\right)=u$. In our construction, the target vector x is used to construct the private keys.

### 2.4. Sampling Algorithms

Our constructions require a vector of form $f=\left(\begin{array}{c} \begin{array}{c} a\\ R^{⊤}a+b \end{array} \end{array}\right)\in R_{q}^{2m}$ where a and b are vectors in $R_{q}^{m}$. Matrix $R\in R^{m\times m}$ consists of polynomials with coefficients $\left\{1,-1\right\}$. We can get the private key by sampling short vectors in $\Lambda_{q}^{u}\left(f\right)$ for some $u\in R_{q}$. Algorithm SampleLeft is used in our construction and algorithm SampleRight is used in our security proof.

Algorithm SampleLeft needs a vector of form $f_{1}:=\left(\begin{array}{c} \begin{array}{c} a\\ m_{1} \end{array} \end{array}\right)$. It inputs a trapdoor $T_{a}$ of $\Lambda_{q}^{⊥}\left(a\right)$ and returns a short vector $s\in \Lambda_{q}^{u}\left(f_{1}\right)$. The description of SampleLeft is shown in Algorithm 1. By algorithm SamplePre and 1, we have $a^{⊤}s_{1}=u-m_{1}^{⊤}s_{2}$. Then, $f_{1}^{⊤}s=a^{⊤}s_{1}+m_{1}^{⊤}s_{2}=u-m_{1}^{⊤}s_{2}+m_{1}^{⊤}s_{2}=u$. Therefore, we get a short vector $s\in R^{m+m_{1}}$ distributed statistical close to $D_{\Lambda_{q}^{u}\left(f_{1}\right),\sigma }$.
**Algorithm 1** SampleLeft$\left(a,m_{1},T_{a},u,\sigma \right)$.**Input:** Polynomial vectors $a\in R_{q}^{m}$ and $m_{1}\in R_{q}^{m_{1}}$, a trapdoor $T_{a}$ of $\Lambda_{q}^{⊥}\left(a\right)$, a polynomial $u\in R_{q}$ and a Gaussian parameter σ;**Output:** A short vector $s\in R_{q}^{m+m_{1}}$ following the Gaussian distribution $D_{\Lambda_{q}^{u}\left(f_{1}\right),\sigma }$ with $f_{1}:=\left(\begin{array}{c} \begin{array}{c} a\\ m_{1} \end{array} \end{array}\right)$.1:Sample a random vector $s_{2}\leftarrow D_{R^{m_{1}},\sigma }$;2:Sample $s_{1}\leftarrow SamplePre\left(a,T_{a},y,\sigma \right)$, where $y=u-m_{1}^{⊤}s_{2}\in R_{q}$;3:**return**$s\leftarrow \left(s_{1},s_{2}\right)\in R^{m+m_{1}}$.

Algorithm SampleRight needs a vector of form $f_{2}:=\left(\begin{array}{c} \begin{array}{c} a\\ R^{⊤}a+b \end{array} \end{array}\right)$. It inputs a trapdoor $T_{b}$ of $\Lambda_{q}^{⊥}\left(b\right)$ and returns a short vector $s\in \Lambda_{q}^{⊥}\left(f_{2}\right)$. The description of SampleRight is shown in Algorithm 2. In HIBE, we also need an algorithm ExtendBasis which is similar to Theorem 1. By algorithm SamplePre and 2, we have $f_{2}s=u$ and then we get a short vector $s\in R_{q}^{m+k}$ distributed statistically close to $D_{\Lambda_{q}^{u}\left(f_{2}\right),\sigma }$.
**Algorithm 2** SampleRight$\left(a,b,T_{b},u,\sigma \right)$.**Input:** Polynomial vectors $a\in R_{q}^{k}$ and $b\in R_{q}^{m}$, a matrix of polynomial $R\in R_{q}^{k\times m}$, a trapdoor $T_{b}$ of $\Lambda_{q}^{⊥}\left(b\right)$, a polynomial $u\in R_{q}$ and a Gaussian parameter σ;**Output:** A short vector $s\in R_{q}^{m+k}$ following the Gaussian distribution $D_{\Lambda_{q}^{u}\left(f_{2}\right),\sigma }$ with $f_{2}:=\left(\begin{array}{c} a\\ R^{⊤}a+b \end{array}\right)$.
1:Select m+k linearly indepndent vectors in $\Lambda_{q}^{⊥}\left(f_{2}\right)$ and construct $T_{f_{2}}$;2:Convert $T_{f_{2}}$ into a basis $T_{f_{2}}^{\prime }$ of $\Lambda_{q}^{⊥}\left(f_{2}\right)$ where $∥\tilde{T_{f_{2}}}∥=∥\tilde{T_{f_{2}}^{\prime }}∥$;3:Sample $s\leftarrow SamplePre(f_{2},T_{f_{2}}^{\prime },u,\sigma )$;4:**return**$s\in \Lambda_{q}^{u}\left(f_{2}\right)$.


## 3. Adaptively Secure IBE

Agrawal [18] converted their selectively secure IBE to an adaptively secure IBE using the technique of Waters [5]. Though the private key size and ciphertext size are the same, the size of the public parameters is too large. In this section, we construct an adaptively security IBE over ideal lattice and reduce the size of the public parameters using the blocking technique.

### 3.1. The IBE Construction

The identity id is an *l* bits string in ${\left\{0,1\right\}}^{l}$. We divide id into $l^{\prime }$ segments $\left(b_{1},b_{2},\cdots ,b_{l^{\prime }}\right)$, where $b_{i}$ is a $l/l^{\prime }=\beta$ bits string. Then, we describe our IBE construction as follows.

**Setup**(λ): On input a security parameter λ and other parameters q,n,m,σ,α, do:Run $(a_{0}$, $T_{a_{0}})$←TrapGen(q,n), where $a_{0}$ is a vector in $R_{q}^{m}$ with a trapdoor $T_{a_{0}}\in R_{q}^{(m-k)\times k}$;Select $l^{\prime }+1$ uniformly random vectors $a_{1},a_{2},\cdots ,a_{l^{\prime }},b\in R_{q}^{m}$, and these vectors are used to form the public parameters;Select a uniformly random polynomial $u\in R_{q}$;Output the public parameters $PP=(a_{0},a_{1},a_{2},\cdots ,a_{l^{\prime }},b,u)$ and master key $MK=(T_{a_{0}})$.

**Extract**(PP,MK,id): On input public parameters PP, master key MK and identity $id=(b_{1},b_{2},\cdots ,b_{l^{\prime }})$, do:Set $a_{id}=b+\sum_{i=1}^{l^{\prime }}b_{i}\cdot a_{i}\in R_{q}^{m}$ and $f=\left(\begin{array}{c} \begin{array}{c} a_{0}\\ a_{id} \end{array} \end{array}\right)\in R_{q}^{2m}$. They are used to generate the private key;Run $s\leftarrow SampleLeft(a_{0},a_{id},T_{a_{0}},u,\sigma )$, where s is a vector in $R_{q}^{2m}$;Output the private key $SK=s\in R_{q}^{2m}$.

**Encrypt**(PP,id,m): On input public parameters PP, an identity $id=(b_{1},b_{2},\cdots ,b_{l^{\prime }})$, and a message $\mu \in {\left\{0,1\right\}}^{n}$, do:Set $a_{id}=b+\sum_{i=1}^{l^{\prime }}b_{i}\cdot a_{i}\in R_{q}^{m}$ and $f=\left(\begin{array}{c} \begin{array}{c} a_{0}\\ a_{id} \end{array} \end{array}\right)\in R_{q}^{2m}$. They are used to generate the ciphertext;Select a uniformly random polynomial $t\in R_{q}$;Select $l^{\prime }$ matrices $R_{1},R_{2},\cdots ,R_{l^{\prime }}$ in $R^{m\times m}$ which consist of uniformly random polynomials with coefficient {1,−1}. Define $R_{id}=\sum_{i=1}^{l^{\prime }}b_{i}R_{i}$ and its coefficients are in $\left\{-l^{\prime }\left(2^{\beta }-1\right),l^{\prime }\left(2^{\beta }-1\right)\right\}$;Select noise polynomial $x\leftarrow D_{R_{q},\sigma }$, noise vector $y\leftarrow D_{R_{q}^{m},\sigma }$ and set $z\leftarrow R_{id}^{⊤}\cdot y\in R_{q}^{m}$;Set $c_{0}=u\cdot t+x+\mu \cdot \left\lfloor q/2\right\rfloor \in R_{q}$, and $c_{1}=f\cdot t+\left[\begin{array}{c} \begin{array}{c} y\\ z \end{array} \end{array}\right]\in R_{q}^{2m}$;Output the ciphertext $CT=\left(c_{0},c_{1}\right)\in R_{q}\times R_{q}^{2m}$.

**Decrypt**(PP,SK,CT): On input public parameters PP, a private key SK=s, and a ciphertext $CT\;=\;(c_{0},c_{1})$, do:Compute $w=c_{0}-s^{⊤}\cdot c_{1}\in R_{q}$, and $w_{i}$ denotes the coefficient of *w*;Compare $w_{i}$ and ⌊q/2⌋ treating them as integer in *Z*, if |w−⌊q/2⌋|<⌊q/4⌋, output 1, otherwise output 0.

### 3.2. Parameters and Correctness

In this section, we prove the correctness of the above IBE scheme. During decryption, we have
(1)$$\begin{array}{cc} w & =c_{0}-s^{⊤}\cdot c_{1}\\ \; & =u\cdot t+x+\mu \cdot \left\lfloor q/2\right\rfloor -s^{⊤}\left(f\cdot t+\left[\begin{array}{c} \begin{array}{c} y\\ z \end{array} \end{array}\right]\right)\\ \; & =\mu \cdot \left\lfloor q/2\right\rfloor +\underset{error\;term}{\underset{︸}{x-s^{⊤}\left[\begin{array}{c} \begin{array}{c} y\\ z \end{array} \end{array}\right]}} \end{array}$$

In order to decrypt correctly, the error term $x-s^{⊤}\left[\begin{array}{c} \begin{array}{c} y\\ z \end{array} \end{array}\right]$ should be bounded by ⌊q/4⌋. Then, we need the following two lemmas to analyze the error rate of decryption.

**Lemma** **1**([41])**.**
*Let c≥1, $C=c\cdot exp(\frac{1-c^{2}}{2})<1$ and $x\leftarrow D_{Z^{n},s}$; then, for any real s>0 and any integer n≥1, we have*
(2)$$Pr\left[∥x∥\geq cs\sqrt{n/2\pi }\right]\leq C^{n}$$

**Lemma** **2**([42])**.**
*For any real s>0, T>0, and any $x\in R^{n}$, we have*
(3)$$Pr\left[\left|<x,D_{Z^{n},s}>\right|\geq Ts∥x∥\right]<2exp\left(-\pi T^{2}\right)$$

**Theorem** **2.**
*Let $q\leq 4\left[l^{\prime }\left(2^{\beta }-1\right)\sqrt{mn}+1\right]\delta c\sigma \sqrt{mn/2\pi }$, c≥1, t>15, the above IBE scheme decrypts correctly with overwhelming probability.*


**Proof** **of** **Theorem** **2.**Letting $s=(\begin{array}{c} \begin{array}{c} s_{1}\\ s_{2} \end{array} \end{array})$ with $s_{1},s_{2}\in R^{m}$, we have $s^{⊤}\left[\begin{array}{c} \begin{array}{c} y\\ z \end{array} \end{array}\right]=s_{1}^{⊤}\cdot y+s_{2}^{⊤}\cdot z$. Since $z=R_{id}^{⊤}\cdot y$, we have $∥z∥=∥R_{id}\cdot y∥\leq ∥R_{id}∥\cdot ∥y∥=l^{\prime }\left(2^{\beta }-1\right)\sqrt{mn}∥y∥$.Similar to [33], we compute the decryption error rate with Lemma 2 as
(4)$$\begin{array}{cc} \; & Pr\left[\left(l^{\prime }\left(2^{\beta }-1\right)\sqrt{mn}+1\right)\sqrt{m}\left|<x,y>\right|\geq q/4\right]\\ \; & =Pr\left[\left|<x,y>\right|\geq q/(4\left(l^{\prime }\left(2^{\beta }-1\right)\sqrt{mn}+1\right)\sqrt{m})\right]\\ \; & =Pr\left[|<x,y>|\geq T\delta ∥x∥\right]\\ \; & <2exp(-\pi T^{2}) \end{array}$$For c≥1, we have $∥x∥\leq c\sigma \sqrt{n/2\pi }$ with Lemma 1. Then,
(5)$$T=\frac{q}{4\left[l^{\prime }\left(2^{\beta }-1\right)\sqrt{mn}+1\right]\sqrt{m}\delta ∥x∥}\geq \frac{q}{4\left[l^{\prime }\left(2^{\beta }-1\right)\sqrt{mn}+1\right]\delta c\sigma \sqrt{mn/2\pi }}$$When *T* is sufficiently large, the decryption error rate $2exp(-\pi T^{2})$ is a negligible function, and we can decrypt correctly with overwhelming probability. □

Similar to [18,19,37], we need to set the parameters as follows:the error term is less than q/4$\left(i.e.\;q\leq 4\left[l^{\prime }\left(2^{\beta }-1\right)\sqrt{mn}+1\right]\delta c\sigma \sqrt{mn/2\pi }\right)$,that algorithm TrapGen can operate (i.e.m=O(nlogq)),that σ is sufficiently large for sampling algorithm(i.e., $\;\sigma >∥\tilde{T_{B}}∥2^{\beta }l^{\prime }\sqrt{m}\omega \sqrt{\log m}=2^{\beta }l^{\prime }\sqrt{m}\omega \sqrt{\log m}$),that reduction applies (i.e., the number of private key queries $Q\leq \frac{q}{2}$).

### 3.3. Security Proof

In this section, we give the security proof of our IBE scheme. We describe the definition of abort-resistant hash functions in Definition 2.

**Lemma** **3.**
*Let q be a prime, the hash family $H_{Wat}$ is $\left(Q,\frac{1}{q}\left(1-\frac{Q}{q}\right),\frac{1}{q}\right)$ abort-resistant where 0<Q<q.*


**Proof** **of** **Lemma** **3.**Let $\bar{id}$ be a set of $\left(id_{0},id_{1},\cdots ,id_{Q}\right)$ where $id_{0}\notin \left\{id_{1},\cdots ,id_{Q}\right\}$. For i=0,⋯,Q+1, $S_{i}$ denotes the set of functions $H(id_{i})=0$ in $H_{Wat}$. We have $|S_{i}|=q^{l^{\prime }-1}$ and $|S_{0}\cap S_{j}|\leq q^{l^{\prime }-2}$ with j>0. For i=1,⋯,Q, the set of $H(id_{0})=0$ and $H(id_{i})\neq 0$ is defined as $S:=S_{0}\backslash \left(S_{1}\cup \cdots \cup S_{Q}\right)$. Then, we have
$$\left|S|=\right|S_{0}\backslash \left(S_{1}\cup \cdots \cup S_{Q}\right)\left|\geq \right|S_{0}|-\sum_{i=1}^{Q}\left|S_{0}\cap S_{i}\right|\geq q^{l^{\prime }-1}-Qq^{l^{\prime }-2}$$
The non-abort probability of $\bar{id}$ is $\left|S\right|/q^{l^{\prime }}\geq \frac{1}{q}\left(1-\frac{Q}{q}\right)$. Since $\left|S|\leq \right|S_{0}|$, the no-abort probability is $\left|S\right|/q^{l^{\prime }}\leq \left|S_{0}\right|/q^{l^{\prime }}\leq \frac{1}{q}$ at most. □

**Theorem** **3.**
*The IBE system with parameters (n,m,q,σ) is IND-ID-CPA secure in the standard model under the hardness of RLWE.*


**Proof** **of** **Theorem** **3.**The proof proceeds in a sequence of games, and the first game is the same as the security game in Definition 1. In game *i*, we use $W_{i}$ to denote that the adversary guesses the challenge message correctly. Then, the advantage of adversary in game *i* is $|Pr\left[W_{i}\right]-\frac{1}{2}|$.**Game 0**. The original IND-ID-CPA game between an adversary A and a challenger.**Game 1**. The challenger builds the public parameters $PP=(a_{0},a_{1},a_{2},\cdots ,a_{l^{\prime }},b,u)$ in the original game. These vectors $a_{1},a_{2},\cdots ,a_{l^{\prime }},b$ are chosen uniformly from $R_{q}^{m}$. The Game 1 challenger chooses $l^{\prime }$ random matrices $R_{i}^{*}\in R^{m\times m}$ and random polynomials $h_{i}\in Z_{q}$ at the setup phase. Matrix $R_{i}^{*}$ consists of uniformly random polynomials with coefficient {−1.1}. Then the challenger generates vectors $a_{0}$ and b as in original game, and constructs vector $a_{i}$ as
$$a_{i}\leftarrow {\left(R_{i}^{*}\right)}^{⊤}\cdot a_{0}-h_{i}\cdot b\in R_{q}^{m},\;i\in \left[1,l^{\prime }\right]$$
The matrix $R_{i}^{*}$ is used to build vector $a_{i}$ and challenge ciphertext $CT^{*}$ (i.e. $z\leftarrow {\left(R_{id}^{*}\right)}^{⊤}y\in R_{q}^{m}$ where $R_{id}^{*}=\Sigma_{i=1}^{l^{\prime }}b_{i}^{*}\cdot R_{i}^{*}\in R^{m\times m}$). Set $R^{*}:=\left(R_{1}^{*},R_{2}^{*},\cdots ,R_{l^{\prime }}^{*}\right)$, the distributions
$$(a_{0}\;,\;a_{0}^{⊤}\cdot R^{*}\;,\;{\left(R^{*}\right)}^{⊤}y)\mathrm{and}(a_{0}\;,\;({\left(a_{1}^{\prime }\right)}^{⊤}\left|\cdots \right|{\left(a_{l^{\prime }}^{\prime }\right)}^{⊤})\;,\;{\left(R^{*}\right)}^{⊤}y)$$
are statistically close. The vectors $a_{i}^{\prime }$ are uniformly random elements in $R_{q}^{m}$. For $z\leftarrow {\left(R_{id}^{*}\right)}^{⊤}\cdot y$, the distributions
$$\left(a_{0}\;,\;a_{0}^{⊤}\cdot R_{1}^{*}\;,\cdots ,\;a_{0}^{⊤}\cdot R_{l^{\prime }}^{*}\;,\;z)\mathrm{and}(a_{0}\;,\;{\left(a_{1}^{\prime }\right)}^{⊤}\;,\cdots ,\;{\left(a_{l^{\prime }}^{\prime }\right)}^{⊤}\;,\;z\right)$$
are statistically close. In adversary’s view, the vectors $a_{0}^{⊤}\cdot R_{i}^{*}$ are statistically close to uniformly random elements ${\left(a_{i}^{\prime }\right)}^{⊤}$ and independent of vector z. Therefore, in adversary’s view, the vector $a_{i}$ are uniformly random vectors as in Game 0. This shows that
(6)$$Pr\left[W_{0}\right]=Pr\left[W_{1}\right]$$**Game 2**. In Game 2, we add an abort event and the rest is the same as Game 1. We use the abort-resistant $H_{Wat}$ introduced in Lemma 3. In the Setup phase, the challenger chooses a function $H\in H_{Wat}$ and reserves it to itself. Then, the challenger answers key queries and sends challenge ciphertext to adversary as in Game 1. We use $id_{1},\cdots ,id_{Q}$ to denote the identities that the adversary queries. We use $id^{*}$ to denote the challenge identity which is not in $\left\{id_{1},\cdots ,id_{Q}\right\}$. In the Guess phase, the adversary returns a guess $r^{\prime }\in \left\{0,1\right\}$. Then, the challenger performs as follows:
**Abort check** [18]: For i=1,⋯,Q, the game proceeds normally if $H(id^{*})=0$ and $H(id_{i})\neq 0$. Otherwise, it resets $r^{\prime }$ and aborts the game. However, the game proceeds normally in the adversary’s view.**Artificial abort** [5,18]: The challenger chooses a bit Γ∈{0,1} such that Pr[Γ=1]=γ(I). If there is no abort γ(I)=0, otherwise, γ(I)=1. If Γ=1 or γ(I)=1, the challenger resets $r^{\prime }$ and aborts the game.For identities $I=(id^{*},id_{1},\cdots ,id_{Q})$, we use ϵ(I) to denote the probability of non-abort when the adversary performs these private key queries. Moreover, we use $\epsilon_{max}$ and $\epsilon_{min}$ to denote the maximum and minimum of ϵ(I).**Lemma** **4**([18])**.**
*For i=1,2, let $W_{i}$ be the event that the adversary wins the Game i. Then,*
$$\left|\mathrm{Pr}\left[W_{2}\right]-\frac{1}{2}\right|\geq \epsilon_{\min}\left|\mathrm{Pr}\left[W_{1}\right]-\frac{1}{2}\right|-\frac{1}{2}\left(\epsilon_{\max}-\epsilon_{\min}\right)$$
According to [18], they show that $\epsilon_{max}-\epsilon_{min}$ is less than $\epsilon_{\min}\left|\mathrm{Pr}\left[W_{1}\right]-\frac{1}{2}\right|$. Since q≥2Q, we have $\epsilon_{min}=\frac{1}{q}\left(1-\frac{Q}{q}\right)\geq \frac{1}{2q}$. Then,
(7)$$\left|\mathrm{Pr}\left[W_{2}\right]-\frac{1}{2}\right|\geq \frac{1}{2}\epsilon_{\min}\left|\mathrm{Pr}\left[W_{1}\right]-\frac{1}{2}\right|\geq \frac{1}{4q}\left|\mathrm{Pr}\left[W_{1}\right]-\frac{1}{2}\right|$$**Game 3**. In Game 3, we change the method of generating $a_{0}$ and b in PP. Vector $a_{0}$ is generated as a random element in $R_{q}^{m}$ and vector b is generated by algorithm TrapGen. The challenger also gets a trapdoor $T_{b}$ of $\Lambda_{q}^{⊥}\left(b\right)$. The construction $a_{i}\leftarrow {\left(R_{i}^{*}\right)}^{⊤}\cdot a_{0}-h_{i}\cdot b\in R_{q}^{m}$ is the same as in Game 2. To answer the private key query of $id=(b_{1},b_{2},\cdots ,b_{l^{\prime }})$, the challenger generates the corresponding private key $SK_{id}=s$ from $\Lambda_{q}^{u}\left(f_{id}\right)$. Let
(8)$$f_{id}:=\left(\begin{array}{c} a_{0}\\ b+\Sigma_{i=1}^{l^{\prime }}b_{i}\cdot a_{i} \end{array}\right)=\left(\begin{array}{c} a_{0}\\ {\left(R_{id}\right)}^{⊤}\cdot a_{0}-h_{id}\cdot b \end{array}\right)$$
where $R_{id}=\Sigma_{i=1}^{l^{\prime }}b_{i}\cdot R_{i}^{*}\in R_{q}^{m\times m}$ and $h_{id}=1+\Sigma_{i=1}^{l^{\prime }}b_{i}\cdot h_{i}\in Z_{q}$. If $h_{id}=0$, the challenger abort the game as in Game 2. Otherwise, the challenger gets $s\leftarrow SampleRight\left(a_{0},h_{id}\cdot b,R_{id},T_{b},u,\sigma \right)\in R_{q}^{2m}$. Then, it sends $SK_{id}=s$ to adversary A.In adversary’s view, Game 2 and Game 3 are indistinguishable. Therefore,
(9)$$Pr\left[W_{2}\right]=Pr\left[W_{3}\right]$$**Game 4**. The challenge ciphertext $\left(c_{0}^{*},c_{1}^{*}\right)$ is randomly selected in $R_{q}\times R_{q}^{2m}$ and the rest is the same as in Game 3, so the advantage of A is 0 in Game 4. Then, we need to prove that Game 3 and Game 4 are computationally indistinguishable.Suppose there is an adversary A who has non-negligible probability in distinguishing Game 3 and Game 4. Then, we constructs an RLWE algorithm B.An instance of RLWE problem is provided as a sample oracle O. We use $O_{\$}$ to denote a truly random oracle. For a random $s\in R_{q}$, we use $O_{s}$ to denote a noisy pseudo-random oracle.**Instance**. For i=0,⋯,m, B requests from O and gets RLWE samples $\left(u_{i},v_{i}\right)\in R_{q}\times R_{q}$ .**Setup**. B generates the public parameters:
Construct random vector $a_{0}\in R_{q}^{m}$ with RLWE samples. For i=1,⋯,m, the *i*-th column of $a_{0}$ is $u_{i}$.Let the random polynomial $u_{0}\in R_{q}$ be the 0-th RLWE sample.Construct vectors $a_{i}$ and b as in Game 3.Send public parameters $PP=(a_{0},a_{1},\cdots ,a_{l^{\prime }},b,u_{0})$ to adversary A.**Phase 1 and Phase 2**. B answers private key queries as in Game 3.**Challenge**. A submits a target identity $id^{*}=\left(b_{1},\cdots ,b_{l^{\prime }}\right)$ and a message $\mu^{*}\in {\left\{0,1\right\}}^{n}$. B prepares a challenge ciphertext for the target identity as follows:
Set $v_{*}=\left[\begin{array}{c} \begin{array}{c} v_{1}\\ \vdots \\ v_{m} \end{array} \end{array}\right]\in R_{q}^{m}$ with the RLWE instance.Let $c_{0}^{*}=v_{0}+\mu^{*}\cdot \left\lfloor q/2\right\rceil \in R_{q}$ to blind the message bit.Set $R_{id}^{*}=\Sigma_{i=1}^{l^{\prime }}b_{i}\cdot R_{i}^{*}\in R_{q}^{m\times m}$ and $c_{1}^{*}=\left[\begin{array}{c} v^{*}\\ {\left(R_{id^{*}}\right)}^{⊤}\cdot v^{*} \end{array}\right]\in R_{q}^{2m}$.Choose a random bit r∈{0,1}. If r=0, set $CT^{*}=\left(c_{0}^{*},c_{1}^{*}\right)$. Otherwise, select a random element $CT^{*}=\left(c_{0},c_{1}\right)$ in $R_{q}\times R_{q}^{2m}$. Then, send challenge ciphertext $CT^{*}$ to adversary.**Guess**. Finally, the adversary A returns a guess $r^{\prime }$. The simulator B outputs 1 if $r^{\prime }=r$, otherwise 0.**Analysis**. According to [18], the challenge ciphertext is the same as valid ciphertext in game 3 if sampling oracle O is pseudo-random $O_{s}$, and the challenge ciphertext is the same as random ciphertext in game 4 if oracle O is truly random $O_{\$}$. The simulator’s advantage in solving RLWE problem is equal to A’s advantage in distinguishing valid ciphertext and random ciphertext. For $Pr\left[W_{4}\right]=\frac{1}{2}$, we get
(10)$$|Pr\left[W_{3}\right]-\frac{1}{2}|=|Pr\left[W_{3}\right]-Pr\left[W_{4}\right]|\leq Adv_{B}^{RLWE}$$Then, we have
(11)$$|Pr\left[W_{0}\right]-\frac{1}{2}|\leq 4q\cdot Adv_{B}^{RLWE}$$ □

## 4. Adaptively Secure HIBE

We extend our IBE scheme to a hierarchical IBE scheme. Similar to our IBE scheme above, we also use the blocking technique to reduce the size of public parameters.

### 4.1. The HIBE Construction

The identity $id|id_{l}$ is composed of *l* identities $id_{i}$ at different depth, and it is represented as $id|id_{l}=\left(id_{1},\cdots ,id_{l}\right)$ where $id_{i}$ is a $l^{\prime }$ bit string. We divide the identity $id_{i}$ at depth *i* into $l^{\prime \prime }$ segments $\left(b_{i,1},\cdots ,b_{i,l^{\prime \prime }}\right)$ where $b_{i,j}$ is a $\beta =l^{\prime }/l^{\prime \prime }$ bits string.

Then, we describe our HIBE construction as follows.

**Setup**(d,λ): On input a security parameter λ, a maximum depth *d* and other parameters q,n,m,σ,α, do:Run $(a_{0}$, $T_{a_{0}})$←TrapGen(q,n), where $a_{0}$ is a vector in $R_{q}^{m}$ with a trapdoor $T_{a_{0}}\in R_{q}^{(m-k)\times k}$;Choose $l^{\prime \prime }d+1$ random vectors $a_{1,1},\cdots ,a_{1,l^{\prime \prime }},\cdots ,a_{d,1},\cdots ,a_{d,l^{\prime \prime }},b\in R_{q}^{m}$, and these vectors are used to form the public parameters;Choose a uniformly random polynomial $u\in R_{q}$;Output the public parameters $PP=(a_{0},a_{1,1},\cdots ,a_{1,l^{\prime \prime }},\cdots ,a_{d,1},\cdots ,a_{d,l^{\prime \prime }},b,u)$ and master key $MK=(T_{a_{0}})$.

**Derive**$\left(PP,id|id_{l},SK_{id|id_{l-1}}\right)$: On input public parameters PP, an identity $id|id_{l}$ and a private key $SK_{id|id_{l-1}}$ at depth l−1, do:Set $f_{id|id_{l}}=\left(\begin{array}{c} f_{id|id_{l-1}}\\ \sum_{i=1}^{l^{\prime \prime }}a_{l,i}b_{l,i}+b \end{array}\right)\in R_{q}^{(l+1)m}$, and it is used to generate the private key;Run $s\leftarrow SampleLeft(f_{id|id_{l-1}},\sum_{i=1}^{l^{\prime \prime }}a_{l,i}b_{l,i}+b,SK_{id|id_{l-1}},\sigma_{l})$, where s is a vector in $R_{q}^{2m}$;Output the private key $SK_{id|id_{l}}=s\in R_{q}^{2m}$.

**Encrypt**(PP,id,m): On input public parameters PP, an identity $id|id_{l}$ at depth *l* and a message $\mu \in {\left\{0,1\right\}}^{n}$, do:Set $f_{id|id_{l}}=\left(\begin{array}{c} f_{id|id_{l-1}}\\ \sum_{i=1}^{l^{\prime \prime }}a_{l,i}b_{l,i}+b \end{array}\right)\in R_{q}^{(l+1)m}$, and it is used to generate the ciphertext;Choose a uniformly random polynomial $t\in R_{q}$;Choose $ll^{\prime \prime }$ matrices $R_{i,j}\in R^{m\times m}$ for i=1,⋯,l and $j=1,\cdots ,l^{\prime \prime }$, which consist of random polynomials with coefficient {1,−1}. Define $R_{id}=\sum_{i=1}^{l^{\prime \prime }}b_{1,i}R_{1,i}\left||\cdots |\right|\sum_{i=1}^{l^{\prime \prime }}b_{l,i}R_{l,i}\in R^{m\times lm}$;Choose noise polynomial $x\leftarrow D_{R_{q},\sigma }$, noise vector $y\leftarrow D_{R_{q}^{m},\sigma }$, and set $z\leftarrow R_{id}^{⊤}\cdot y\in R_{q}^{lm}$;Set $c_{0}=u\cdot t+x+\mu \cdot \left\lfloor q/2\right\rfloor \in R_{q}$, and $c_{1}=f\cdot t+\left[\begin{array}{c} \begin{array}{c} y\\ z \end{array} \end{array}\right]\in R_{q}^{(l+1)m}$;Output the ciphertext $CT=\left(c_{0},c_{1}\right)\in R_{q}\times R_{q}^{(l+1)m}$.

**Decrypt**$\left(PP,SK_{id|id_{l}},CT\right)$: On input public parameters PP, a private key $SK_{id|id_{l}}$ at depth *l* and a ciphertext $CT=(c_{0},c_{1})$, do:Set $\tau_{l}:=\sigma_{l}\sqrt{m(l+1)}w\sqrt{\log(lm)}$;Sample $s_{id}\leftarrow SamplePre\left(f_{id|id_{l}},SK_{id|id_{l}},u,\tau_{l}\right)$ such that $f_{id}\cdot s_{id}=u$;Compute $w=c_{0}-s_{id}^{⊤}\cdot c_{1}\in R_{q}$, $w_{i}$ denotes the coefficient of *w*;Compare $w_{i}$ and ⌊q/2⌋ treating them as integer in *Z*, if $|w_{i}-\left\lfloor q/2\right\rfloor |<\left\lfloor q/4\right\rfloor$, output 1, otherwise output 0.

### 4.2. Parameters and Correctness

In this section, we prove the correctness of the above HIBE scheme. During decryption, we have
(12)$$\begin{array}{cc} w & =c_{0}-s_{id}^{⊤}\cdot c_{1}\\ \; & =u\cdot t+x+\mu \cdot \left\lfloor q/2\right\rfloor -s_{id}^{⊤}\left(f\cdot t+\left[\begin{array}{c} \begin{array}{c} y\\ z \end{array} \end{array}\right]\right)\\ \; & =\mu \cdot \left\lfloor q/2\right\rfloor +\underset{error\;term}{\underset{︸}{x-s_{id}^{⊤}\left[\begin{array}{c} \begin{array}{c} y\\ z \end{array} \end{array}\right]}} \end{array}$$

In order to decrypt correctly, the error term $x-s_{id}^{⊤}\left[\begin{array}{c} \begin{array}{c} y\\ z \end{array} \end{array}\right]$ should be bounded by ⌊q/4⌋. Similar to our IBE scheme, the following proof also needs Lemmas 1 and 2 to analyze the error rate of decryption.

**Theorem** **4.**
*Let $q\leq 4\left[l^{\prime \prime }\left(2^{\beta }-1\right)\sqrt{lmn}+1\right]\delta c\sigma \sqrt{mn/2\pi },c\geq 1,t>15$, the above HIBE scheme decrypts correctly with overwhelming probability.*


**Proof** **of** **Theorem** **4.**Letting $s_{id}=\left(\begin{array}{c} \begin{array}{c} s_{1}\\ s_{2} \end{array} \end{array}\right)$ with $s_{1},s_{2}\in R^{m}$ we have $s_{id}^{⊤}\left[\begin{array}{c} \begin{array}{c} y\\ z \end{array} \end{array}\right]=s_{1}^{⊤}\cdot y+s_{2}^{⊤}\cdot z$. Since $z=R_{id}^{⊤}\cdot y$, we have $∥z∥=∥R_{id}\cdot y∥\leq ∥R_{id}∥\cdot ∥y∥=l^{\prime \prime }\left(2^{\beta }-1\right)\sqrt{lmn}∥y∥$.Then, we compute the decryption error rate with Lemma 2 as
(13)$$\begin{array}{cc} \; & Pr\left[\left(l^{\prime \prime }\left(2^{\beta }-1\right)\sqrt{lmn}+1\right)\sqrt{m}\left|<x,y>\right|\geq q/4\right]\\ \; & =Pr\left[\left|<x,y>\right|\geq q/(4\left(l^{\prime \prime }\left(2^{\beta }-1\right)\sqrt{lmn}+1\right)\sqrt{m})\right]\\ \; & =Pr\left[|<x,y>|\geq T\delta ∥x∥\right]\\ \; & <2exp(-\pi T^{2}) \end{array}$$For c≥1, we have $∥x∥\leq c\sigma \sqrt{n/2\pi }$ with Lemma 1. Then
(14)$$T=\frac{q}{4\left[l^{\prime \prime }\left(2^{\beta }-1\right)\sqrt{lmn}+1\right]\sqrt{m}\delta ∥x∥}\geq \frac{q}{4\left[l^{\prime \prime }\left(2^{\beta }-1\right)\sqrt{lmn}+1\right]\delta c\sigma \sqrt{mn/2\pi }}$$When *T* gets large enough, the decryption error rate $2exp(-\pi T^{2})$ is negligible, and we can decrypt correctly with overwhelming probability.  □

Similar to [18,19,37], we need to set the parameters as follows:the error term is less than q/4$\left(i.e.,\;q\leq 4\left[l^{\prime \prime }\left(2^{\beta }-1\right)\sqrt{lmn}+1\right]\delta c\sigma \sqrt{mn/2\pi }\right)$,that algorithm TrapGen can operate (i.e.,m=O(nlogq)),that σ is sufficiently large for sampling algorithm(i.e., $\;\sigma >∥\tilde{T_{B}}∥2^{\beta }l^{\prime \prime }\sqrt{l}m\omega \sqrt{\log m}=2^{\beta }l^{\prime \prime }\sqrt{l}m\omega \sqrt{\log m})$,that reduction applies (i.e., the number of private key queries $Q\leq q^{l}/2$).

### 4.3. Security Proof

In this section, we give the security proof of our HIBE scheme. We describe the definition of abort-resistant hash functions in Definition 2.

**Lemma** **5.**
*Let q be a prime and 0<Q<q; the hash family $H_{Wat}$ is $\left(Q,\frac{1}{q^{l}}\left(1-\frac{Q}{q^{l}}\right),\frac{1}{q^{l}}\right)$ abort-resistant.*


**Proof** **of** **Lemma** **5.**Let $\bar{id}$ be a set of $\left(id_{0},id_{1},\cdots ,id_{Q}\right)$ where $id_{0}\notin \left\{id_{1},\cdots ,id_{Q}\right\}$. For i=0,⋯,Q+1, we have $|S_{i}|=q^{l(l^{\prime \prime }-1)}$ and $|S_{0}\cap S_{j}|\leq q^{l(l^{\prime \prime }-2)}$ for j>0. Then,
(15)$$\left|S|=\right|S_{0}\backslash \left(S_{1}\cup \cdots \cup S_{Q}\right)\left|\geq \right|S_{0}|-\sum_{i=1}^{Q}\left|S_{0}\cap S_{i}\right|\geq q^{l(l^{\prime \prime }-1)}-Qq^{l(l^{\prime \prime }-2)}$$The non-abort probability of $\bar{id}$ is $\left|S\right|/q^{ll^{\prime \prime }}\geq \frac{1}{q^{l}}\left(1-\frac{Q}{q^{l}}\right)$. Since $\left|S|\leq \right|S_{0}|$, the non-abort probability is $\left|S\right|/q^{ll^{\prime \prime }}\leq \left|S_{0}\right|/q^{ll^{\prime \prime }}\leq \frac{1}{q^{l}}$ at most. □

**Theorem** **5.**
*The HIBE system with parameters (n,m,q,σ) is IND-ID-CPA secure for depth d in the standard model under the hardness of RLWE.*


**Proof** **of** **Theorem** **5.**The proof proceeds in a sequence of games, and the first game is the same as the security game in Definition 1. In game *i*, we use $W_{i}$ to denote that adversary guess the challenge message correctly. The advantage of adversary in game *i* is $|Pr\left[W_{i}\right]-\frac{1}{2}|$.**Game 0**. The original IND-ID-CPA game between an adversary A and a challenger.**Game 1**. The challenger builds the public parameters $PP=(a_{0},a_{1,1},\cdots ,a_{1,l^{\prime \prime }},\cdots ,a_{d,l^{\prime \prime }},b,u)$ in the original game. These vectors $a_{1,1},\cdots ,a_{1,l^{\prime \prime }},\cdots ,a_{d,l^{\prime \prime }},b$ are chosen uniformly random from $R_{q}^{m}$.The Game 1 challenger chooses $ll^{\prime \prime }$ random matrices $R_{k,i}^{*}\in R^{m\times m}$ and polynomials $h_{k,i}\in R_{q}$ for $k\in \left[1,l\right],i\in \left[1,l^{\prime \prime }\right]$. Matrix $R_{k,i}^{*}$ consists of uniformly random polynomials with coefficients {−1.1}. Then, the challenger generates vectors $a_{0}$ and b as in original game, and constructs vector $a_{k,i}$ as
(16)$$a_{k,i}\leftarrow {\left(R_{k,i}\right)}^{⊤}\cdot a_{0}-h_{k,i}\cdot b\in R_{q}^{m},k\in \left[1,l\right],i\in \left[1,l^{\prime \prime }\right]$$In the adversary’s view, the distribution $a_{0}^{⊤}\cdot R_{k,i}^{*}$ is statistically close to uniform ${\left(a_{k,i}^{\prime }\right)}^{⊤}$ and independent of vector z. Therefore, in adversary’s view, vecors $a_{k,i}$ are uniformly random elements as in Game 0. This shows that
(17)$$Pr\left[W_{0}\right]=Pr\left[W_{1}\right]$$**Game 2**. In Game 2, we add an abort event which is similar to the abort event in Section 3.3. The rest is the same as Game 1. We use the abort-resistant $H_{Wat}$ introduced in Lemma 5.According to [18], they show that $\epsilon_{max}-\epsilon_{min}$ is less than $\epsilon_{\min}\left|\mathrm{Pr}\left[W_{1}\right]-\frac{1}{2}\right|$. Since $q^{l}\geq 2Q$, we have $\epsilon_{min}=\frac{1}{q^{l}}\left(1-\frac{Q}{q^{l}}\right)\geq \frac{1}{2q^{l}}$. By Lemma 4, we have
(18)$$\left|\mathrm{Pr}\left[W_{2}\right]-\frac{1}{2}\right|\geq \frac{1}{2}\epsilon_{\min}\left|\mathrm{Pr}\left[W_{1}\right]-\frac{1}{2}\right|\geq \frac{1}{4q^{l}}\left|\mathrm{Pr}\left[W_{1}\right]-\frac{1}{2}\right|$$**Game 3**. In Game 3, we change the method of generating $a_{0}$ and b in PP. Vector $a_{0}$ is generated as a random vector in $R_{q}^{m}$ and vector b is generated by algorithm TrapGen. The challenger also gets a trapdoor $T_{b}$ of $\Lambda_{q}^{⊥}\left(b\right)$. The construction $a_{k,i}\leftarrow {\left(R_{k,i}^{*}\right)}^{⊤}\cdot a_{0}-h_{k,i}\cdot b\in R_{q}^{m}$ is the same as in Game 2. To answer the private key query of $id=(id_{1},id_{2},\cdots ,id_{l})$, the challenger generates the corresponding private key $SK_{id}=s$ from $\Lambda_{q}^{u}\left(f_{id}\right)$. Let
(19)$$f_{id|id_{l}}:=\left(\begin{array}{c} a_{0}\\ \sum_{i=1}^{l^{\prime \prime }}a_{1,i}b_{1,i}+b\\ \vdots \\ \sum_{i=1}^{l^{\prime \prime }}a_{l,i}b_{l,i}+b \end{array}\right)\;or\;f_{id}=\left(\begin{array}{c} a_{0}\\ {\left(R_{id}\right)}^{⊤}\cdot a_{0}-h_{id}\cdot b \end{array}\right)$$
where
(20)$$R_{id}:=\sum_{i=1}^{l^{\prime \prime }}b_{1,i}R_{1,i}^{*}\left||\cdots |\right|\sum_{i=1}^{l^{\prime \prime }}b_{l,i}R_{l,i}^{*}\in R^{m\times lm}$$
and
(21)$$h_{id}=\left(1+\sum_{i=2}^{l^{\prime \prime }}b_{1,i}\cdot h_{1,i}\right)\left||\cdots |\right|\left(1+\sum_{i=1}^{l^{\prime \prime }}b_{l,i}\cdot h_{l,i}\right)$$If $h_{id}=0$, the challenger aborts the game as in Game 2. Otherwise, the challenger gets private key $s\leftarrow SampleRight\left(a_{0},h_{id}\cdot b,R_{id},T_{b},u,\sigma \right)\in R_{q}^{2m}$. Then, it sends $SK_{id}=s$ to the adversary A. In the adversary’s view, Game 2 and Game 3 are indistinguishable. Therefore,
(22)$$Pr\left[W_{2}\right]=Pr\left[W_{3}\right]$$**Game 4**. The challenge ciphertext $\left(c_{0}^{*},c_{1}^{*}\right)$ is randomly selected in $R_{q}\times R_{q}^{2m}$ and the rest is the same as in Game 3, so the advantage of A is 0 in Game 4. Similar to Section 3.3, we need to prove that Game 3 and Game 4 are computationally indistinguishable.**Instance**. For i=0,⋯,m, B receives RLWE samples $\left(u_{i},v_{i}\right)\in R_{q}\times R_{q}$.**Setup**. B generates the public parameters:
Construct random vector $a_{0}\in R_{q}^{m}$ with RLWE samples. For i=1,⋯,m, the *i*-th column of $a_{0}$ is $u_{i}$.Let a random polynomial $u_{0}\in R_{q}$ be the 0-th RLWE sample.Construct $a_{k,i}$ and b as in Game 3.Send public parameters $PP=(a_{0},a_{1,1},\cdots ,a_{1,l^{\prime \prime }},\cdots ,a_{d,l^{\prime \prime }},b,u)$ to adversary A.**Phase 1 and Phase 2**. B answers private key queries as in Game 3.**Challenge**. A submits a target identity $id^{*}=\left(id_{1}^{*},\cdots ,id_{l}^{*}\right)$ and a message $\mu^{*}\in {\left\{0,1\right\}}^{n}$. B returns a challenge ciphertext as follows:
Set $v^{*}=\left[\begin{array}{c} \begin{array}{c} v_{1}\\ \vdots \\ v_{m} \end{array} \end{array}\right]\in R_{q}^{m}$ with the RLWE instance.Set $c_{0}^{*}=v_{0}+\mu^{*}\cdot \left\lfloor q/2\right\rceil \in R_{q}$ to blind the message bit.Set $R_{id^{*}}:=\sum_{i=1}^{l^{\prime \prime }}b_{1,i}R_{1,i}^{*}\left||\cdots |\right|\sum_{i=1}^{l^{\prime \prime }}b_{l,i}R_{l,i}^{*}$ and $c_{1}^{*}=\left[\begin{array}{c} v^{*}\\ {\left(R_{id^{*}}^{*}\right)}^{⊤}\cdot v^{*} \end{array}\right]$.Choose a random bit r∈{0,1}. If r=0 set $CT^{*}=\left(c_{0}^{*},c_{1}^{*}\right)$, otherwise, select a random $CT^{*}=\left(c_{0},c_{1}\right)$ in $R_{q}\times R_{q}^{2m}$. Then, send the challenge ciphertext $CT^{*}$ to adversary.**Guess**. Finally, the adversary A returns a guess $r^{\prime }$. The simulator B outputs 1 if $r^{\prime }=r$ otherwise 0.**Analysis**. According to [18], the challenge ciphertext is the same as valid ciphertext in game 3 if sampling oracle O is pseudo-random $O_{s}$, and the challenge ciphertext is the same as random ciphertext in game 4 if oracle O is truly random $O_{\$}$. The simulator’s advantage in solving RLWE problem is equal to A’s advantage in distinguishing valid ciphertext and random ciphertext. For $Pr\left[W_{4}\right]=\frac{1}{2}$, we get
(23)$$|Pr\left[W_{3}\right]-\frac{1}{2}|=|Pr\left[W_{3}\right]-Pr\left[W_{4}\right]|\leq Adv_{B}^{RLWE}$$Then
(24)$$|Pr\left[W_{0}\right]-\frac{1}{2}|\leq 4q^{l}\cdot Adv_{B}^{RLWE}$$ □

## 5. Efficiency

Trade-off. We make a trade-off between the decrease in the size of public parameters and the increase in the computation cost. Using the blocking technique, we divide an identity into $l^{\prime }$ segments, and the number of elements in public parameters is reduced from l+2 to l/β+2 where β is a flexible constant. Therefore, the percentage of decrease in public parameter space is $\frac{l-l^{\prime }}{l+2}$ and it is shown as the thin blue line in Figure 1 with l=160. According to the analysis of Singh [19], there is no effect of $l^{\prime }$ on cost of key generation, encryption and decription. However, we need to increase the value of lattice modulo *q* for maintaining the same security level, and it will increase the computation cost. According to Chatterjee ’s work [20], the number of bits in *q* is increased by $\Delta =\beta -\log_{2}\beta$. We use |q| to denote the bit length of *q* and then $|q^{\prime }|=|q|+2\Delta =|q|+2\left(\beta -\log_{2}\beta \right)$. The percentage of increase in computation cost is $\frac{|q^{\prime }\left|-|q\right|}{\left|q\right|}=\frac{2(\beta -\log_{2}\beta )}{\left|q\right|}$ and it is shown as the thick red line in Figure 1 with |q|=256. In Figure 1, the *x*-axis represents the value of β, and the *y*-axis represents the percentage of increase or decrease. For l=160 and |q|=256, the size of public parameters is reduced by 89.7% while the cost of computation is merely increased by 5.2% when $l^{\prime }=16$ or β=10. If we set $l^{\prime }=8$ or β=20, the size of public parameters is reduced by 93.8% while the computational cost is merely increased by 12.25%.

Comparisons. We propose an adaptively secure IBE scheme in Section 3. Table 1 shows the comparison of storage space between different IBE schemes in the standard model. In this table, PP, SK, *l* denote the public parameters, private keys and length of user’s identity.

Since the public parameters are composed of multiple matrices, its size will directly affect the communication overhead in actual applications. As shown in this table, the public parameter in Agrawal’s construction [18] contains l+2 matrices. Zhang’s construction [23] achieves shorter public parameter at the cost of weaker security guarantees. In Yamada’s construction [22], the public parameter consists of $d\lceil l^{1/d}\rceil +2$ matrices, where *d* is a constant. In Katsumata’s scheme [36], the public parameter consists of $d\lceil l^{1/d}\rceil +2$ vectors because of ring setting. The relationship between the size of public parameters and constant *d* is shown in Figure 2. For l=160, the minimum size of public parameters is 17 vectors when we set d=5. Moreover, we need to set *d* very small (e.g., d=2 or 3) because of the reduction cost. If we set d=2 (resp. 3), the public parameters have 28 (resp. 20) vectors. In [24], the public parameter consists of $\log^{2}l+2$ matrices via new partitioning functions. In our construction, the public parameters only contain $l^{\prime }+2$ vectors, where $l^{\prime }=l/\beta$. We have analyzed the choice of β or $l^{\prime }$ in the previous part. For l=160, the public parameter only contains 10 (resp. 18) vectors if we choose β=20 (resp. 10).

The comparison of public parameter size is shown in the Figure 3. It involves four IBE schemes with short public parameters, including Yam17 [24], KY16 [36] (d=3), ZCZ16 [23] and ours (β=20). The *x*-axis represents the length of user’s identity, and the *y*-axis represents the number of basic matrices (or vectors) in the public parameters of each scheme. Obviously, the public parameters in our scheme are shorter than [24] and [36]. Moreover, it can be shorter than [23] if the identity length *l* is small (e.g., less than 140).

Compared with the LWE-based scheme, the RLWE-based scheme contains a lot of polynomial operations instead of matrix operations. To compare more fair, we only compare the computational efficiency between the schemes under RLWE assumption. Since the scheme by [36] also has short public parameters and ring setting, we only compare the calculation efficiency between [36] and our scheme. Table 2 shows the comparison of computational efficiency. In this table, KeyGen, Enc, Dec denote the key generation, encryption and decryption.

The difference between these two schemes is the calculation of H(id) and $a_{id}$. In Katsumata’s construction [36], $H\left(id\right)=b+\Sigma_{j_{1},\cdots ,j_{d}}PubEval_{d}\left(b_{1,j_{1}},b_{2,j_{2}},\cdots ,b_{d,j_{d}}\right)$ and it is used to generate private keys. They use the homomorphic function $PubEval_{d}:{\left(R_{q}^{m}\right)}^{d}\to R_{q}^{m}$ as in [22], which maps vectors $b_{1},\cdots ,b_{d}$ to a vector in $R_{q}^{m}$. The function PubEval needs $dm^{2}n^{2}$ multiplications and d−1 inversions. In our construction, $a_{id}=b+\Sigma_{i=1}^{l^{\prime }}b_{i}\cdot a_{i}$ and it is also used as the input of the sampling algorithm to generate private keys. However, it only needs $l^{\prime }mn$ multiplication operations which is obviously less than [36].

In Section 4, we also extend our IBE scheme to an adaptively secure HIBE scheme. Using Waters’ technology, we can convert the selectively secure HIBE scheme to adaptive security. Howerve, the size of the public parameter increases from d+2 matrices to $dl^{\prime }+2$ matrices. In our HIBE construction, the public parameter is reduced from $dl^{\prime }+2$ matrices to $dl^{\prime \prime }+2$ vectors where $l^{\prime \prime }=l^{\prime }/\beta$. In particular, it can be further reduced to $l^{\prime \prime }+2$ thanks to the method of Chatterjee [11,43]. Finally, both of our constructions support multi-bit encryption because of ring setting.

## 6. Conclusions

In this paper, we propose an identity-based encryption scheme and a hierarchical identity-based encryption scheme over ideal lattice. The new schemes have short public parameters, and achieve IND-ID-CPA security in the standard model. In addition, we use the trapdoor of Micciancio to further improve the efficiency of our scheme. However, there are still many problems to be solved, such as how to reduce the size of ciphertext and how to implement these schemes.

## Figures and Tables

**Figure 1 entropy-22-01247-f001:**
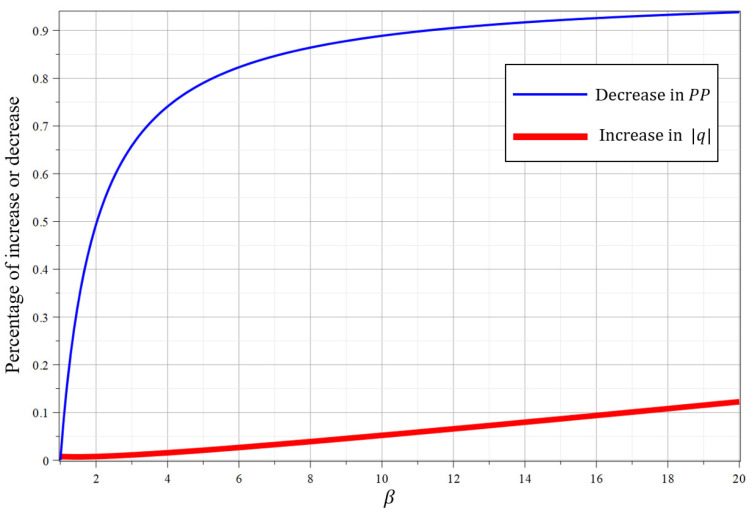
Relative decrease in PP and relative increase in |q|.

**Figure 2 entropy-22-01247-f002:**
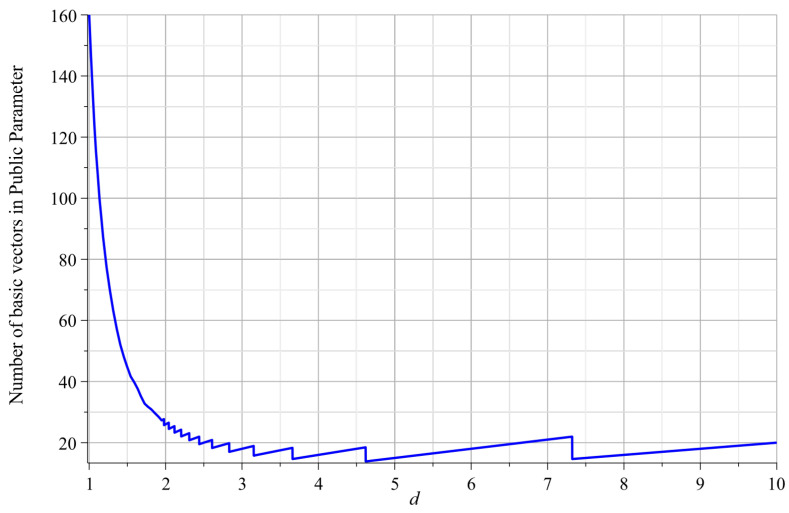
The relationship between the size of public parameters and constant *d*.

**Figure 3 entropy-22-01247-f003:**
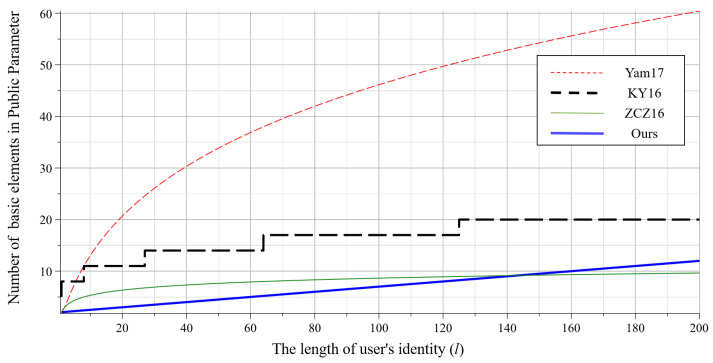
Comparison of public parameter size in different schemes.

**Table 1 entropy-22-01247-t001:** Comparison of storage space.

Schemes	PP Size	SK Size	Ciphertext Size	Security	Assumption
[18]	(l+2)mnlogq	2mlogq	(2m+1)logq	Adaptive-CPA	LWE
[23]	(logl+2)mnlogq	mnlogq	(m+n)logq	Adaptive-CPA	LWE
[22] *	$(d\left\lceil l^{1/d}\right\rceil +2)mn\log q$	2mlogq	(2m+1)logq	Adaptive-CPA	LWE
[36] *	$(d\left\lceil l^{1/d}\right\rceil +2)mn\log q$	2mnlogq	(2m+1)nlogq	Adaptive-CPA	RLWE ^†^
[24]	$(\log^{2}l+2)mn\log q$	2mlogq	(2m+1)logq	Adaptive-CPA	LWE
Ours **	(l/β+2)mnlogq	2mnlogq	(2m+1)nlogq	Adaptive-CPA	RLWE ^†^

* In [22] and [36], they use an injective map which maps an identity $id\in {\left\{0,1\right\}}^{l}$ to a subset of ${\left[1,\left\lceil l^{1/d}\right\rceil \right]}^{d}$, where the element *d* is a flexible constant. The choice of *d* will affect the reduction cost; ** In our construction, the element β is a flexible constant. The choice of β will affect the size of modulus *q* and we make a trade-off in the previous part; † Our scheme and [36] only work over the rings $R_{q}$; thus, the basic elements in the public parameters are polynomial vectors rather than matrices.

**Table 2 entropy-22-01247-t002:** Comparison of computational efficiency.

Schemes	KeyGen	Enc	Dec
[36]	$dm^{2}n^{2}$	$dm^{2}n^{2}+n^{2}+2mn$	$2mn^{2}$
Ours	$l^{\prime }mn$	$l^{\prime }mn+n^{2}+2mn$	$2mn^{2}$

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
