# Peer review of "Adaptively Secure Efficient (H)IBE over Ideal Lattice with Short Parameters"

_entropy, 2020, doi:10.3390/e22111247_

Round 1
Reviewer 1 Report
All my comments on the previous submission were met. I believe the paper is in good shape and can be published. The contribution was made clear.
Author Response
Thanks for your all suggestions, and we have learned a lot from them.
Reviewer 2 Report
This paper appears to be well constructed and provide a useful contribution to the development of (H)IBE schemes. The paper is generally well written and well contrasted with the state of the art. Overall I recommend this paper for acceptance, some minor improvements are suggested below, but apart for typos/grammar these are not necessary for acceptance by this reviewer.
The most difficult aspect of this paper for a reader is the density of the writing. Sections 3 & 4 are both approximately 4 pages of dense text and mathematics with limited explanation or discussion. This could be made easier and clearer to read with the addition of some minor text to explain the path the reader is following.
The efficiency discussion (Section 5) appears to not compare the computational efficiency with other schemes (Figure 1), this may be interesting for readers who wish to compare the authors' work with those in Table 1 for efficiency. Similarly Table 2 does not address all the schemes discussed above.
Lastly, the English has some very small mistakes and errors. Some details are below. Fixing these would be nice, but the English does not prevent understanding of the paper.
Line 6: "of the most" -> "of most"
Lines 13-14: This is clearer if the "when l' is set of 16 (resp. 8)" before the other parts (i.e. reduction and computation). This also occurs on lines 72-73
Line 15: "proved achieving" -> "proved to achieve"
Line 24: "It can be used", is it here: IBE, PKG, or something else?
Line 26: "under bilinear" -> "under the bilinear"
Line 29: "is inefficient to be practical" -> "is too inefficient to be practical"
Line 34: "can not" -> "cannot"
Lines 83-85: no space before open brackets (multiple cases)
Author Response
Point 1: The most difficult aspect of this paper for a reader is the density of the writing. Sections 3 & 4 are both approximately 4 pages of dense text and mathematics with limited explanation or discussion. This could be made easier and clearer to read with the addition of some minor text to explain the path the reader is following.
Response 1: Thanks for this comment. In the revised paper, we have added some short explanations to make the article easier to understand.
Point 2: The efficiency discussion (Section 5) appears to not compare the computational efficiency with other schemes (Figure 1), this may be interesting for readers who wish to compare the authors' work with those in Table 1 for efficiency. Similarly Table 2 does not address all the schemes discussed above.
Response 2: Thanks for this comment. Figure 1 shows the trade-off between the storage cost and the computation cost of our IBE scheme. The value of may directly affect the computation cost and security. By Figure 1, we can find a appropriate parameter to ensure the efficiency and security of our scheme.
In Table 1, we compare the storage cost of LWE-based and RLWE-based IBE schemes. Compared with the LWE-based schemes, the RLWE-based schemes contain a lot of polynomial operations instead of matrix operations. In order to compare more fair, we only compare the computational efficiency between the schemes under RLWE assumption in Table 2 (i.e., [KY16] and ours).
Point 3: Lastly, the English has some very small mistakes and errors. Some details are below. Fixing these would be nice, but the English does not prevent understanding of the paper.
Response 3: Thank you very much for pointing out the mistakes in our paper, and we have corrected these writing issues.

This manuscript is a resubmission of an earlier submission. The following is a list of the peer review reports and author responses from that submission.
Round 1
Reviewer 1 Report
This paper proposes an adaptively secure (H)IBE based on the RLWE assumption in the standard model. The proposed constructions provides shorter public parameters than previous ones.
Comments:
1) In 2014, a method was proposed in [SRB14] to construct an adaptively secure HIBE scheme (based on the LWE assumption) with short public parameters using the Chatterjee and Sarkar’s blocking technique. In this paper, it is difficult to find a meaning beyond replacing the technique of [SRB14] from LWE-based to RLWE-based, and other than this, the specific contribution of this paper is not properly described.
2) In the Comparison of Section 5, the proposed scheme is mainly compared with the one proposed in [KY16]. First of all, there is much room for improvement in the description. More detailed and accurate analysis results should be presented to get a better understanding of the contribution claimed in this paper.
3) Since [KY16], more efficient adaptively secure HIBE schemes based on the LWE assumption, such as [Yam17], have been proposed. It seems that these constructions can also be converted to a ring version (i.e., over ideal lattice). It is essential to mention the conversion of the most efficient LWE-based scheme proposed so far to that over ideal lattice.
4) There are other minor typos and editing corrections. It would be better to supplement these parts so that the paper has a higher quality.
This paper is well structured, but it lacks freshness of idea and has a high dependency on existing result in [SRB14]. It is difficult to find contributions other than applying the techniques of [SRB14] to the scheme over ideal lattice, and this paper does not fully describe the distinct contributions. In addition, there is a strong need to improve the description of the introduction (including related work) and performance analysis sections. Therefore, this paper is not sufficient to be accepted for publication in this journal.
[SRB14] K. Singh, C.P. Rangan, and A.K. Banerjee, “Efficient Lattice HIBE in the Standard Model with Shorter Public Parameters,” in ICT-EurAsia 2014, LNCS 8407, pp. 542-553, 2014.
[KY16] S. Katsumata and S. Yamada, “Partitioning via Non-linear Polynomial Functions: More Compact IBEs from Ideal Lattices and Bilinear Maps,” in ASIACRYPT 2016, Part II, LNCS 10032, pp. 682-712, 2016.
[Yam17] S. Yamada, “Asymptotically Compact Adaptively Secure Lattice IBEs and Verifiable Random Functions via Generalized Partitioning Techniques,” in CRYPTO 2017, Part III, LNCS 10403, pp. 161-193, 2017.
Reviewer 2 Report
Please see the attached pdf.
